# Gas Sensitivity of IBSD Deposited TiO2 Thin Films

Aleksei V. Almaev [1,2,*], Nikita N. Yakovlev [1], Bogdan O. Kushnarev [1], Viktor V. Kopyev [1], Vadim A. Novikov [1], Mikhail M. Zinoviev [1,3,4], Nikolay N. Yudin [1,3,4], Sergey N. Podzivalov [1,3], Nadezhda N. Erzakova [1], Andrei V. Chikiryaka [5], Mikhail P. Shcheglov [5], Houssain Baalbaki [1] and Alexey S. Olshukov [1]

[1] Research and Development Center for Advanced Technologies in Microelectronics, National Research Tomsk State University, 634050 Tomsk, Russia
[2] Fokon LLC, 248035 Kaluga, Russia
[3] Laboratory of Optical Crystals LLC, 634040 Tomsk, Russia
[4] Vladimir Zuev Institute of Atmospheric Optics, Russian Academy of Sciences, Siberian Branch, 634055 Tomsk, Russia
[5] Ioffe Institute of the Russian Academy of Sciences, 194021 Saint Petersburg, Russia
* Correspondence: almaev_alex@mail.ru

**Abstract:** TiO2 films of 130 nm and 463 nm in thickness were deposited by ion beam sputter deposition (IBSD), followed by annealing at temperatures of 800 °C and 1000 °C. The effect of $H_2$, CO, $CO_2$, $NO_2$, NO, $CH_4$ and $O_2$ on the electrically conductive properties of annealed TiO2 thin films in the operating temperature range of 200–750 °C were studied. The prospects of IBSD deposited TiO2 thin films in the development of high operating temperature and high stability $O_2$ sensors were investigated. TiO2 films with a thickness of 130 nm and annealed at 800 °C demonstrated the highest response to $O_2$, of 7.5 arb.un. when exposed to 40 vol. %. An increase in the annealing temperature of up to 1000 °C at the same film thickness made it possible to reduce the response and recovery by 2 times, due to changes in the microstructure of the film surface. The films demonstrated high sensitivity to $H_2$ and nitrogen oxides at an operating temperature of 600 °C. The possibility of controlling the responses to different gases by varying the conditions of their annealing and thicknesses was shown. A feasible mechanism for the sensory effect in the IBSD TiO2 thin films was proposed and discussed.

**Keywords:** ion beam sputter deposition (IBSD); TiO2; gas sensor

## 1. Introduction

Titanium oxide (TiO2) belongs to the class of wide-gap semiconductors with intrinsic *n*-type conductivity [1–7]. Gas sensors [1–3,8–10], photodetectors and solar panels [11,12], memristors [13] and photocatalysts [14,15] have been developed. based on TiO2 films, due to their having structural, optical, electrically conductive and catalytic properties, high thermal and chemical stability, being relatively cheap, with good availability. The metastable anatase and brookite phases of TiO2 at the temperatures of 600–1000 °C transform into the stable rutile phase [9]. All three phases of TiO2 exhibit sensitivity to gases, depending on the method of their preparation, and their electrical, structural and dimensional parameters [1–3,8–10]. Resistive [1–3,8–10] and capacitive [16,17] sensors based on TiO2 are being developed, including for high operating temperature applications. The resistive sensors are easy to implement and relatively cheap. In many cases, they consist of a semiconductor film on the surface of an insulating substrate with metal contacts. The resistive sensor also includes a heater to stimulate physical and chemical processes between the film surface and gas molecules.

TiO2 thin films are highly sensitive to gases due to an increase in the contribution of surface conductance, which largely depends on the charge state of the film surface [2,3,10,18]. One of the techniques to produce thin films of metal oxide semiconductors is physical vapor deposition (PVD), including thermal evaporation (EV), pulsed laser deposition (PLD), magnetron sputtering (MS), and ion beam sputter deposition (IBSD). These methods are

used to deposit high-quality thin films of metal oxide semiconductors for various applications, with a wide range of thicknesses, to combine the production of thin films with microelectronic technologies, and to modify the composition of, and control the properties of, films by varying the conditions of their deposition [19–28]. IBSD is distinguished by the highest energies of forming particles. as well as a large number of parameters affecting the deposition process and the possibility of achieving a higher vacuum in the operating mode [27–34]. This allows fine varying of the electrically conductive, structural, optical, mechanical, and other properties of the films during deposition, and to deposit more uniform layers in thickness and composition over large areas of the substrates. The IBSD thin films are characterized by better adhesion, denser structure, fewer defects, close to ideal stoichiometry and higher purity, compared to films deposited by other PVD methods.

The gas-sensitive properties of metal oxide semiconductor thin films fabricated by the IBSD method (see Table 1) have been much less studied than those obtained by the MS and PLD. In Table 1, $d$ is the film thickness; $n_g$ is the target gas concentration; $T_{MAX}$ is the operating temperature of maximum response; and $S$ is the response to gas. The ratio $R_{air}/R_g$ was chosen as the film's response to the gas, where $R_{air}$ is the film's resistance in pure air and $R_g$ is the film resistance in a mixture of air + target gas. The thicknesses of the $In_2O_3$, $SnO_2$, and $MoO_3$ films did not exceed 500 nm and for most films $d$ were below 100 nm. The films described above were characterized by the absence of pores and grain structure. but demonstrated a high response to low concentrations of reducing gases [35–38]. In ref [38] it was shown that the response to 0.01 vol. % $NH_3$ of $MoO_3$ films, deposited by IBSD with a less developed surface, was 2.5 times higher than the response of films obtained by the sol–gel method. The IBSD was also used to deposit layers of catalyst metals on film surfaces with a thickness of a few nanometers. This made it possible to significantly increase $S$ and reduce $T_{MAX}$ [33,37]. The optimal value of $d$ for the $SnO_2$ film thickness was 100 nm [35]. The sensitivity mechanism of the IBSD synthesized films without a grain structure is rather similar to the sensitivity mechanism of single-crystalline metal oxide semiconductor thin film [39].

**Table 1.** Gas sensitive characteristics of IBSD metal oxide semiconductors (MOS) thin films.

| MOS | $d$ (nm) | Gas | $n_g$ (vol. %) | $T_{MAX}$ (°C) | S | Ref. |
|---|---|---|---|---|---|---|
| $In_2O_3$ | | | | 432 | 1.02 | |
| $Au/In_2O_3$ | 40 | Isoprene | $25 \times 10^{-4}$ | 364 | 2.82 | [33] |
| $Pt/In_2O_3$ | | | | 255 | 2.51 | |
| $Pd/In_2O_3$ | | | | 196 | 6.31 | |
| $SnO_2$ | 10 | H$_2$ | 0.1 | 350 | 20 | [35] |
| | 20 | | | | 25 | |
| | 50 | | | | 60 | |
| | 100 | | | | 70 | |
| | 200 | | | | 30 | |
| | 500 | | | | 20 | |
| $SnO_2$ | 50 | $C_4H_{10}$ | 0.5 | 400 | 4 | [36] |
| $Pt/SnO_2$ | | | | | 4.5 | |
| $SnO_2$:Ca | | | | | 2.5 | |
| $Pt/SnO_2$:Ca | | | | | 2.3 | |
| $SnO_2$ | 50 | $CH_4$ | 0.5 | 400 | 1.75 | [37] |
| $SnO_2$:Ca | | | | | 1.4 | |
| $MoO_3$ | - | $NH_3$ | 0.01 | 450 | 4 | [38] |

Meanwhile, there are no publications devoted to the gas sensitive properties of IBSD $TiO_2$ thin films. Refs [29–32] investigated of the impact of IBSD parameters on the optical, structural, and mechanical properties of $TiO_2$ thin films. Argon, oxygen, and xenon ions were used to sputter the Ti and $TiO_2$ targets. It was shown that the properties of the films were weakly affected by the type of targets, energy, and type and incidence angle of ions. However, the scattering geometry had a significant effect on the film's properties.

Our present research is devoted to a comprehensive study on the structural, electrical, and, most of all, gas-sensitive properties of pure $TiO_2$ thin films, obtained by the IBSD technique.

## 2. Experimental Methods

$TiO_2$ thin films were fabricated by the IBSD technique using Aspira-200 equipment with an annular beam of ion source. The sputtered target was a 5-inches Ti disk, with a purity of 99.995 wt. %. The diameter of the ion beam focused on the target was ~25 mm. Ar (99.995 vol. %) and $O_2$ (99.7 vol. %) were used as the working gases. The ratio of partial flows $Ar/O_2$ was $\frac{1}{2}$ at a total flow of 30 cm$^3$/min. Polished polycrystalline sapphire plates were chosen as substrates. Prior to deposition of $TiO_2$ films, the substrates were cleaned using high purity acetone. and subsequently washed in bi-distilled water. The substrates were cleaned by means of an auxiliary ion source with a source power of ~40 W and an ion energy of ~150 eV for 10 min before film deposition. The substrate temperature during film growth was kept at 100 °C. The films were deposited at a gas pressure in the chamber of $5 \times 10^{-4}$ Pa. The thicknesses of the $TiO_2$ films were 130 nm and 463 nm. The average deposition rate of the $TiO_2$ films was 0.3 Å/s. After sputtering, the $TiO_2$ thin films were annealed at $T_{ann}$ = 800 °C and 1000 °C in air for 60 min. As-deposited $TiO_2$ films are amorphous [28]. $TiO_2$ films are of interest for high-temperature operating gas sensors with $T > 600$ °C [40]. Annealing temperatures of $T_{ann}$ = 800 °C and 1000 °C are known to prevent changes in the microstructure of the films during high temperature heating. Pt contacts were deposited on the $TiO_2$ film surfaces through a mask to measure the gas sensitive properties.

The surface morphology of the films was studied by atomic force microscopy (AFM). X-ray diffraction analysis (XRD) was performed to determine the phase composition of the films. The XRD measurements were carried out using a diffractometer with CuK$\alpha$ radiation operated at 40 kV and 30 mA. The X-ray source wavelength was 1.54 Å. Transmission spectra, in the wavelength range of $\lambda$ = 310–485 nm, were studied for $TiO_2$ films deposited on single-crystalline sapphire substrates.

The current–voltage (*I–V*) characteristics and time dependences of the sample resistance under exposure to various gases were measured by a Keithley 2636A source-meter and a hermetic Nextron MPS-CHH micro-probe station. The measurements were carried out under dark conditions and in a flow of dry pure air, or in a gas mixture of dry pure air + target gas. $H_2$, CO, $CO_2$, $NO_2$, NO, $CH_4$ and $O_2$ were selected as target gases. A mixture of $N_2$ and $O_2$ was used to study the film sensitivity to oxygen. The flow rate of gas mixtures through the measurement chamber was maintained at 1000 cm$^3$/min. The source of dry pure air was a special generator. The concentration of the target gas in the mixture was controlled by a gas mixture generator with a Bronkhorst gas mass flow controller. The relative error of the gas flow rate did not exceed 1.5%. The samples were mounted on a hot stage and heated to the desired operating temperature, $T_{oper}$, in the range from RT to 750 °C, where RT was the room temperature. The $T_{oper}$ was controlled by the Nextron MPS-CHTC controller. The accuracy of the $T_{oper}$ setting was ±0.1 °C. The applied voltage *U* to the samples was 5 V.

## 3. Results and Discussion

### 3.1. Structural Properties of TiO$_2$ Films

The surface morphology of the as-deposited films consisted of grains, which, after annealing, formed large agglomerates (illustrated in Figure 1). The presence of a grain structure for films without annealing was caused by the low quality of the polycor substrate,

which was considered appropriate to use for the manufacture of cheap sensor chips. The surface morphology parameters of the $TiO_2$ thin films are compared in Table 2, where $D_a$ was the size of $TiO_2$ agglomerates along the substrate plane and $D_g$ was the $TiO_2$ grain size along the substrate plane. An increase in $d$ and $T_{ann}$ led to an increase in $D_a$ and $D_g$.

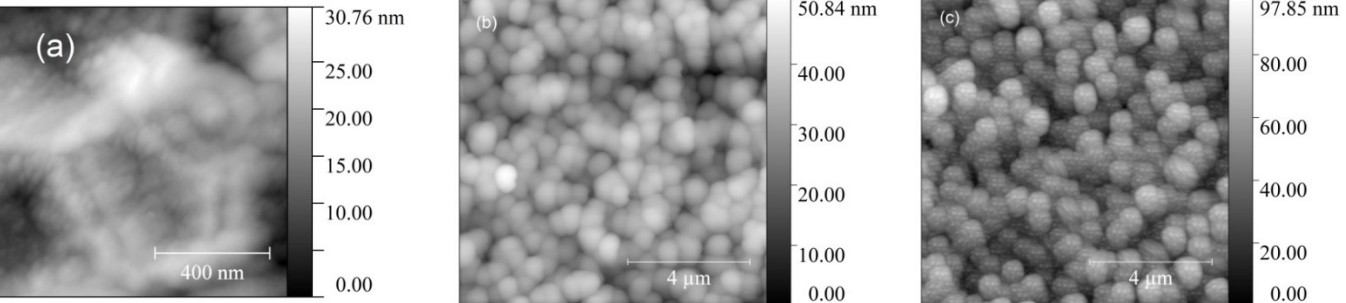

**Figure 1.** AFM images of $TiO_2$ thin films surface at $d$ = 463 nm and $T_{ann}$ = 0 (**a**), $T_{ann}$ = 800 °C (**b**); $T_{ann}$ = 1000 °C (**c**).

**Table 2.** Surface microrelief parameters of $TiO_2$ thin films.

| $d$ (nm) | $T_{ann}$ (°C) | $D_a$ (nm) | $D_g$ (nm) |
|---|---|---|---|
| | 0 | - | 10–30 |
| 130 | 800 | 320–550 | 100 |
| | 1000 | 500–700 | 125 |
| | 0 | - | 10–30 |
| 463 | 800 | 350–600 | 100–110 |
| | 1000 | 600–800 | 150 |

The XRD spectra of the as-prepared films, exhibited in Figure 2, demonstrated many low-intensity peaks that could be associated with different crystallographic planes of the corundum $Al_2O_3$ phase. The as-deposited films were amorphous and the XRD spectra of these films corresponded to polycrystalline sapphire substrate. After annealing, the positions of the $Al_2O_3$ peaks persisted, but the intensity of these peaks decreased. Peaks at $2\theta$ = 39.7° and $2\theta$ = 46.17° on the XRD spectra appeared after annealing. The second peak was associated with the (202) crystallographic plane of the $Al_2O_3$ [41]. The high-intensity peak at $2\theta$ = 39.7° could be associated with the (200) crystallographic plane of the rutile $TiO_2$ phase. The position of this peak did not depend on $T_{ann}$. There was a slight shift in the positions of the peaks to the right, probably due to the presence of elastic deformations in the films after high-temperature annealing [42]. The XRD spectra of IBSD $TiO_2$ thin films differed sharply from the spectra of $TiO_2$ films and nanosized structures obtained by other methods [14,43–46]. The intense peak at $2\theta$ = 38.65°, associated with the (200) crystallographic plane of the $TiO_2$ rutile phase, was observed for MS deposited $TiO_2$ thin films annealed at $T_{ann}$ = 900 °C [47]. The authors were unable to find data on the study of the XRD spectra of the IBSD deposited $TiO_2$ thin films.

Direct optical transitions take place for $TiO_2$ films, regardless of $T_{ann}$, that is characteristic for the rutile phase [48,49]. An increase in $T_{ann}$ from 800 °C to 1000 °C led to a decrease in the energy of band gap $E_g$ from 3.5 eV to 3.2 eV. The value $E_g$ = 3.2 $eV$ is typical for the rutile $TiO_2$ phase [48,49]. The higher value of $E_g$ at $T_{ann}$ = 800 °C could be explained by the existence of a small fraction of the anatase crystalline phase with a larger $E_g$ in the $TiO_2$ film's structure.

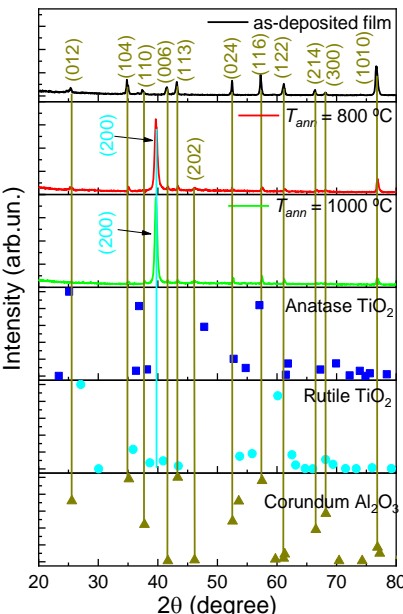

**Figure 2.** XRD spectra of TiO$_2$ films annealed at $T_{ann}$ = 800 °C and 1000 °C.

Thus, it could be concluded that the annealed films composed of the rutile phase of TiO$_2$. TiO$_2$ films annealed at $T_{ann}$ = 800 °C might contain a small amount of the anatase modification. Increasing $T_{ann}$ up to 1000 °C led to the final transition of the TiO$_2$ films into the high-temperature rutile phase.

### 3.2. Electrically Conductive Properties of TiO$_2$ Thin Films in Dry Pure Air

The I–V characteristics of TiO$_2$ thin films with Pt contacts were linear in the ranges of $U$ = 0–20 V, $T_{oper}$ = 200–750 °C and $T_{oper}$ = 400–750 °C after annealing at $T_{ann}$ = 800 °C and 1000 °C, respectively. For TiO$_2$ films annealed at $T_{ann}$ = 800 °C the differential conductance $G_d$ increased exponentially with $T_{oper}$ from 200 °C to 750 °C without any features characteristic of MOS thin films [50]. A decrease in the film thickness from 463 nm to 130 nm led to an increase in $G_d$ by 1–2 orders of magnitude. An increase in $T_{ann}$ up to 1000 °C led to a decrease in $G_d$ by 40 times at $d$ = 130 nm and by 700 times at $d$ = 463 nm. The activation energies of conduction $\Delta E_d$ did not depend on $d$ (Figure 3), at $T_{ann}$ = 800 °C $\Delta E_d$ = 1.29 ± 0.08 eV and at $T_{ann}$ = 1000 °C $\Delta E_d$ = 2.36 ± 0.05 eV.

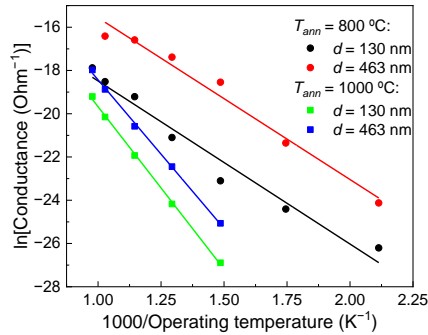

**Figure 3.** Arrhenius curves for TiO$_2$ films after annealing at $T_{ann}$ = 800 °C and 1000 °C.

The $\Delta E_d$ of films annealed at $T_{ann}$ = 800 °C was close to the values for bulk samples obtained by ceramic technology with annealing at 800 °C [51]. The activation energy of films annealed at $T_{ann}$ = 1000 °C coincided with the value of $\Delta E_d$ for single-crystal bulk samples obtained by the Verneuil melt method [52]. Such values of $\Delta E_d$ are typical for

high-temperature treatments, or grown methods, and are caused by the presence of oxygen vacancies in $TiO_2$.

### 3.3. The Effect of Oxygen on the Electrically Conductive Properties of $TiO_2$ Thin Films

$TiO_2$ films are of interest for the development of high operating temperature $O_2$ sensors for extreme environments [53–55], due to their high chemical and thermal stability. The effect of $O_2$ led to a reversible increase in the resistance of $TiO_2$ thin films placed in an atmosphere of dry $N_2$ (Figure 4). The rise of the resistance of the film under exposure to $O_2$, and the drop of resistance after this exposure, were approximated by the following functions, respectively:

$$R(t) = R_{Ost} - A\exp[-t/\tau_1], \tag{1}$$

$$R(t) = R_{Nst} + B\exp[-t/\tau_2], \tag{2}$$

where $R$ is the resistance of a $TiO_2$ thin film, $t$ is time and $R_{Ost}$ and $R_{Nst}$ are the stationary values of the film resistance in the $N_2 + O_2$ mixture and in $N_2$, respectively. $A$ and $B$ are constants; $\tau_1$ and $\tau_2$ are time constants. The operation speed of gas sensors is determined by response $t_{res}$ and recovery $t_{rec}$ times. These values are given by the relaxation times of adsorption $\tau_A$ and desorption $\tau_D$ of gas molecules on the solid surface, $\tau_A$ & $\tau_D \sim \exp[(E_D-E_A)/(2kT)]$, where $E_D$ and $E_A$ are the activation energies of desorption and adsorption processes of gas molecules on the semiconductor surface, $k$ is the Boltzmann constant, and $T$ is the absolute temperature of the semiconductor. The values, $\tau_A$, $\tau_D$, and, hence, $t_{res}$, $t_{rec}$ sharply decrease with $T_{oper}$. The values $\tau_1$ and $\tau_2$ are related to $\tau_A$ and $\tau_D$, respectively. From Equations (1) and (2), the exponents at $t \geq 2.3\tau_1$ and $t \geq 2.3\tau_2$ can be neglected. The resistance of the $TiO_2$ thin film achieved $R_{Ost}$ and $R_{Nst}$ at $t \geq 2.3\tau_1$ and $t \geq 2.3\tau_2$. Thus, $t_{res} = 2.3\tau_1$ and $t_{rec} = 2.3\tau_2$. Estimates of $t_{res}$ and $t_{rec}$ for $TiO_2$ thin films at $T_{oper} = 750\,^\circ C$ are presented in Table 3.

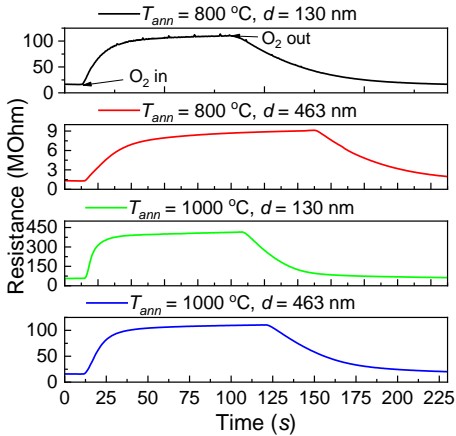

**Figure 4.** Time dependences of the $TiO_2$ thin films resistances upon exposure to 40 vol. % $O_2$ and $T_{oper} = 750\,^\circ C$.

**Table 3.** Response and recovery times of $TiO_2$ thin films upon exposure to 40 vol. % $O_2$ and $T_{oper} = 750\,^\circ C$.

| $T_{ann}$ (°C) | $d$ (nm) | $t_{res}$ (s) | $t_{rec}$ (s) |
|---|---|---|---|
| 800 | 130 | 35.0 | 85.3 |
| | 463 | 54.7 | 85.1 |
| 1000 | 130 | 12.6 | 47.8 |
| | 463 | 23.9 | 92.6 |

Increasing $T_{ann}$ at a fixed value of $d$ and decreasing $d$ at a fixed value of $T_{ann}$ led to a decrease in $t_{res}$. Changing the film thickness at $T_{ann} = 800\,^\circ C$ did not affect $t_{rec}$, but, at $T_{ann} = 1000\,^\circ C$, a decrease in the film's thickness led to a decrease in $t_{rec}$ by about 2 times.

At $d$ = 130 nm, an increase in $T_{ann}$ led to a significant decrease in $t_{rec}$ by 1.8 times, and at $d$ = 463 nm, there was a slight increase in $t_{rec}$. The increase in $t_{res}$ and $t_{rec}$ with increasing $d$ and fixed $T_{ann}$ are explained by the formation of a more developed surface. It slows down the diffusion of oxygen molecules and atoms, on the one hand, and of oxygen vacancies, on the other hand [56]. The decrease in $t_{res}$ with an increase in $T_{ann}$ and a fixed thickness was caused by an increase in the size of grains and agglomerates, which led to the opposite effect. It is worth noting that $t_{res}$ and $t_{rec}$ contained the time required to establish the stationary state of the atmosphere in the measuring chamber, which, according to our estimates, could reach 6 s.

The following ratio, $S_O$, was chosen as the film response to $O_2$:

$$S_O = R_{Ost}/R_{Nst}. \tag{3}$$

Films annealed at $T_{ann}$ = 800 °C showed sensitivity to $O_2$ in the range of $T_{oper}$ = 300–750 °C (shown in Figure 5). For these films the maximum response was observed at $T_{oper}$ = 750 °C. At $T_{oper}$ = 700–750 °C films with $d$ = 130 nm were characterized by the highest responses to $O_2$. In the range of $T_{oper}$ = 300–600 °C $TiO_2$ thin films with a thickness of 463 nm and annealed at $T_{ann}$ = 800 °C demonstrated a slightly higher response to $O_2$.

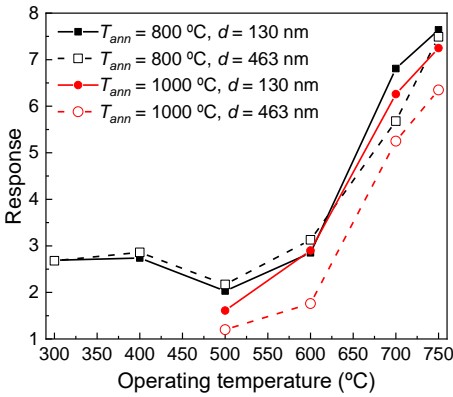

**Figure 5.** Dependences of the response to 40 vol. % $O_2$ on operating temperature.

For $TiO_2$ thin films annealed at $T_{ann}$ = 1000 °C the response to $O_2$ was measured in the range of $T_{oper}$ = 500–750 °C. At $T_{oper}$ = 300–500 °C and these films were not sensitive to $O_2$. An increase in $T_{ann}$ led to a decrease in the response to $O_2$ at fixed values of $d$, due to the effect of changes in the surface microrelief. The films were not sensitive to $O_2$ at $T_{oper}$ = 300–500 °C for the same reason. A decrease in the response at fixed $T_{oper}$ with $d$ was observed, due to an increase in the contribution of bulk conductivity. The maximum responses to $O_2$ for these films within $T_{oper}$ range of 500–750 °C took place at $T_{oper}$ = 750 °C. We believe that the response for these films would increase with a further increase in $T_{oper}$ [1–3,8–10]. Measurements at $T_{oper}$ >750 °C were limited by the capabilities of measuring equipment.

The dependences of the $TiO_2$ thin film responses to the $O_2$ concentration at $T_{oper}$ = 750 °C (displayed in Figure 6) were approximated by the power function $S_O \sim n_{O2}{}^l$, where $n_{O2}$ is the $O_2$ concentration and $l$ is the power index. A change in the film thickness at a fixed value of $T_{ann}$ had little effect on the value of $l$ (Table 4). An increase in $T_{ann}$ led to a decrease in $l$ by ~0.025.

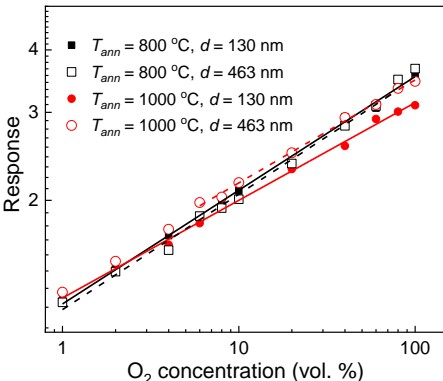

**Figure 6.** Dependences of the response on the $O_2$ concentration at $T_{oper}$ = 750 °C.

**Table 4.** Power indexes *l* for $TiO_2$ thin films at $T_{oper}$ = 750 °C.

| $T_{ann}$ (°C) | $d$ (nm) | $l$ |
|:---:|:---:|:---:|
| 800 | 130 | $0.227 \pm 0.004$ |
| | 463 | $0.235 \pm 0.006$ |
| 1000 | 130 | $0.195 \pm 0.003$ |
| | 463 | $0.212 \pm 0.002$ |

### 3.4. Sensitivity of $TiO_2$ Thin Films to Reducing and Oxidizing Gases

At $T_{oper}$ = 600 °C and 750 °C the responses of $TiO_2$ films to fixed concentrations of reducing and oxidizing gases were measured (Figure 7). The resistance of the films decreased when exposed to reducing gases: CO, $CH_4$, and $H_2$. Under exposure to oxidizing gases, $CO_2$, NO and $NO_2$, the resistance of the films reversibly increased. The responses to reducing gases were determined by the following formula:

$$S = R_{airst}/R_{gst}, \tag{4}$$

where $R_{airst}$ and $R_{gst}$ are stationary values of the $TiO_2$ film resistance in dry pure air and in a mixture of dry pure air + target gas, respectively. The responses to oxidizing gases were determined by the inverse relation of (4).

$TiO_2$ films at $d$ = 463 nm and $T_{ann}$ = 800 °C demonstrated relatively high responses to 1 vol. % $H_2$ and 0.01 vol. % $NO_2$. $TiO_2$ films at $d$ = 130 nm and $T_{ann}$ = 1000 °C demonstrated relatively high responses to 1 vol. % CO, $CO_2$ and $CH_4$. At $T_{oper}$ = 600 °C responses to 40 vol. % $O_2$ were lower than the responses to 1 vol. % $H_2$, regardless of the film thickness and $T_{ann}$, as well as being lower than the responses of $TiO_2$ films at $d$ = 130 nm and $T_{ann}$ = 1000 °C to 1 vol. % CO, $CO_2$ and $CH_4$. Responses of $TiO_2$ films to fixed concentrations of other gases, in comparison with the response to 40 vol. % $O_2$ at different $d$ and $T_{ann}$, were comparable or lower. Increasing the operating temperature of $TiO_2$ thin films up to 750 °C, regardless of $d$ and $T_{ann}$, led to a decrease in responses to all gases, except for $O_2$ and $CH_4$. Responses to $O_2$ and $CH_4$ increased with $T$.

A more significant Increase in the response to $O_2$, as well as a decrease in responses to other gases, should be expected with a further increase in $T_{oper}$. It is believed that at $T_{oper}$ > 750 °C the sensitivity to gases is realized due to the interaction with oxygen vacancies [53–55]. In this operating temperature range the sensors should be selectively sensitive to $O_2$. However, chemisorption of gas molecules on the semiconductor surface and their interaction with semiconductor defects can occur at $T_{oper}$ >750 °C. It often leads to the manifestation of high operating temperature sensitivity to reducing gases. Due to safety requirements and technical limitations the comparison of responses to the same concentration of different gases was not experimentally studied. For this reason, the

sensitivity of $TiO_2$ thin films to oxygen $\beta_O$ and other gases $\beta$ was compared, using the following ratios, respectively:

$$\beta_O = (R_{Ost} - R_{Nst})/n_{O2},$$
$$\beta = (R_{airst} - R_{gst})/n_g \text{ for reducing gases,} \quad (5)$$
$$\beta = (R_{gst} - R_{airst})/n_g \text{ for oxidizing gases.}$$

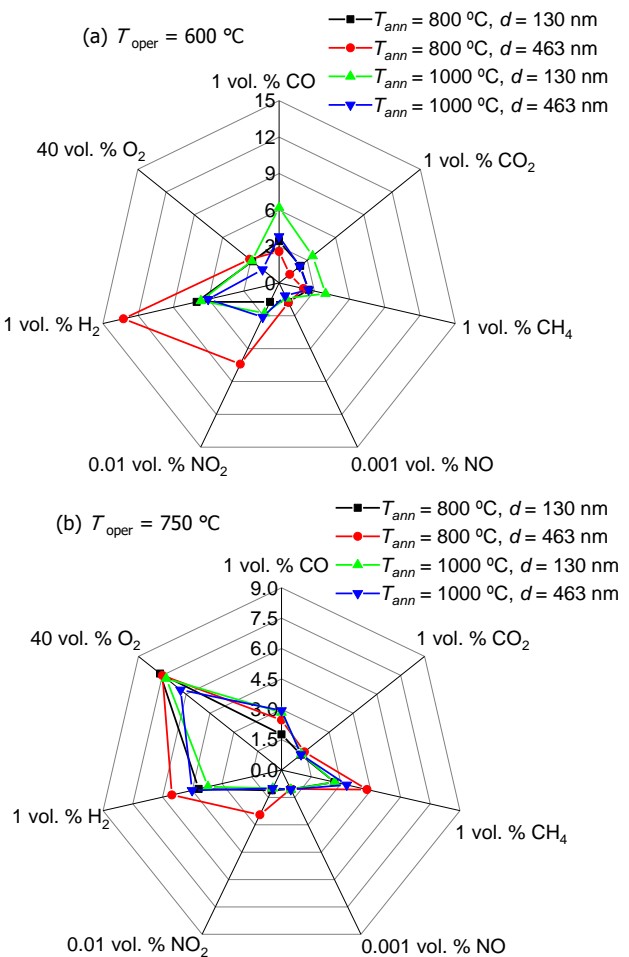

**Figure 7.** Responses to fixed concentrations of various gases at $T_{oper}$ = 600 °C (**a**) and $T_{oper}$ = 750 °C (**b**).

It is worth noting that the estimates of values $\beta_O$ and $\beta$ (Figure 8) did not take into account the possible saturation of the gas-sensitive characteristics of the films under exposure to used gas concentrations. The films were characterized by the highest sensitivity to nitrogen oxides and showed approximately the same sensitivity to CO, $CO_2$, $CH_4$ and $H_2$ at $T_{oper}$ = 600 °C, regardless of $d$ and $T_{ann}$. The lowest sensitivity of $TiO_2$ films was realized when exposed to $O_2$. An increase in $T_{oper}$ up to 750 °C led to a decrease in $\beta_O$ and $\beta$, due to a decrease in the base resistance. In this case, the ratios between the sensitivities to different gases were preserved. $TiO_2$ films at $d$ = 130 nm and $T_{ann}$ = 1000 °C were characterized by the highest sensitivity to gases. $TiO_2$ films at $d$ = 463 nm and $T_{ann}$ = 800 °C demonstrated the lowest sensitivity to gases.

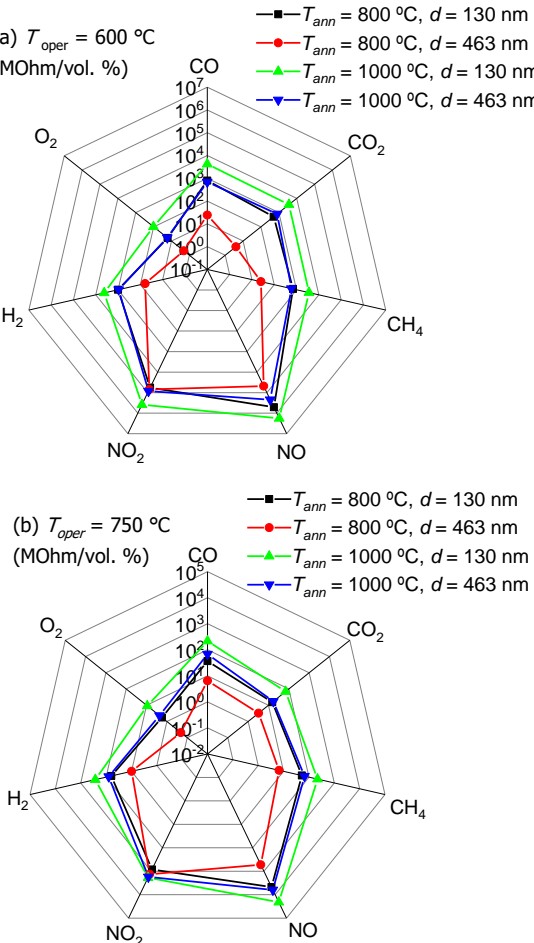

**Figure 8.** Sensitivity of TiO$_2$ thin films to various gases at $T_{oper}$ = 600 °C (**a**) and $T_{oper}$ = 750 °C (**b**).

The repeatability of IBSD TiO$_2$ thin film characteristics was investigated under cyclic exposure to H$_2$ and O$_2$ gases (see Figure 9). Obviously, an increase in the thickness of films at a fixed $T_{ann}$ value led to an improvement in the repeatability of resistance and response of samples when exposed to H$_2$. The standard deviations of $S$ at $d$ = 463 nm were 5% and 2% at $T_{ann}$ = 800 °C and 1000 °C, respectively. Response to H$_2$ for films at $d$ = 130 nm and $T_{ann}$ = 800 °C increased with each cycle of gas exposure, due to an increase in $R_{airst}$. For these films $S$ increased 1.4 times after 6 cycles of H$_2$ exposure. Films at $d$ = 130 nm and $T_{ann}$ = 1000 °C were not characterized by high repeatability of the resistance and $S$ under exposure to H$_2$. The IBSD TiO$_2$ thin films demonstrated high reproducibility of resistance and $S_O$ under cyclic exposure to O$_2$. The standard deviations of $S_O$ when exposed to O$_2$ were 1–2%. The reason for the instability of the IBSD fabricated TiO$_2$ thin film characteristics at $d$ = 130 nm might be the process of Ti reduction by hydrogen. This process was considered in detail for MS deposited SnO$_2$ thin films at $d$ = 100 nm [57]. The density of adsorption centers on the TiO$_2$ film surface increases at Ti reduction by hydrogen. Oxygen is primarily chemisorbed onto these newly formed adsorption centers, which leads to an increase in $R_{airst}$ and $S$. It is worth mentioning that this process became significant with a decrease in the film thickness, as the contribution of surface electrically conductance increased. The contribution of this process became insignificant with increasing $d$.

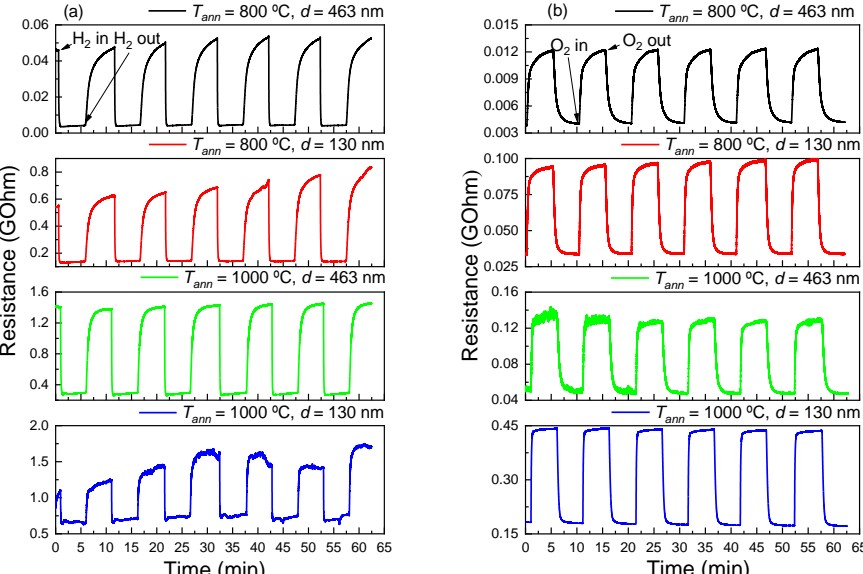

**Figure 9.** Time dependences of the TiO$_2$ thin films resistances upon exposure to 1 vol. % H$_2$ at $T_{oper}$ = 600 °C (**a**) and upon exposure to 40 vol. % O$_2$ at $T_{oper}$ = 750 °C (**b**).

Thus, the IBSD TiO$_2$ thin films demonstrated high sensitivity to H$_2$, NO$_2$ and exhibited potential for the development of O$_2$ sensors operating at high temperatures. For this reason, a comparison of sensitivity to O$_2$, H$_2$ and NO$_2$ for thin films obtained by different CVD and PVD methods is presented in Table 5, where CVD represents chemical vapor deposition. In Table 5, RFMS is radio-frequency magnetron sputtering, DCMS is direct-current magnetron sputtering, MOCVD is metalorganic chemical vapor deposition, EBE + oxidation is electron beam evaporation with following oxidation and ALD is atomic laser deposition.

**Table 5.** Comparison of sensitivity to O$_2$, H$_2$ and NO$_2$ for thin films deposited by different CVD and PVD methods.

| Methods | $d$ (nm) | $n_g$ (vol. %) | $T_{oper}$ (°C) | $S$ | Ref. |
|---------|----------|----------------|-----------------|-----|------|
| O$_2$ | | | | | |
| RFMS | 50 | 0.6 | 500 | 1.14 | [58] |
| DCMS | 60 | 10 | RT | 76 | [59] |
| IBSD | 130 | 40 | 750 | 7.64 | This work |
| H$_2$ | | | | | |
| DCMS | 50 | 1 | 450 | ~$10^4$ | [47] |
| DCMS | 160 | 1 | 120 | $10^7$ | [60] |
| RFMS | ~300 | 1 | 175 | 1.8 | [61] |
| MOCVD | 71 | 1 | 100 | 1.3 | [62] |
| EBE + oxidation | 25 | 0.05 | 500 | 91 | [63] |
| DCMS | 400 | 3 | 150 | $1.1 \times 10^4$ | [64] |
| IBSD | 463 | 1 | 600 | 13.25 | This work |
| NO$_2$ | | | | | |
| RFMS | 10 | 0.05 | RT | 1.021 | [65] |
| ALD | 8 | 0.1 | RT | $7.5 \times 10^{-7}$ * | [66] |
| IBSD | 463 | 0.01 | 600 | 7.41 | This work |

* The response was determined from the frequency properties.

The IBSD deposited TiO$_2$ thin films were characterized by high response to O$_2$ at high operating temperatures. In refs. [58,59] the sensitivity to O$_2$ of much thinner TiO$_2$ films were studied. UV irradiation of films was employed to increase the response at RT [59]. At the same time, $t_{res}$ was 360 s, which gave rise to significantly larger $t_{res}$ and $t_{rec}$ for the IBSD TiO$_2$ thin films (see Table 3).

The response to H$_2$ of IBSD deposited TiO$_2$ thin films was not high in comparison with other samples. The advantage of IBSD deposited TiO$_2$ thin films was relatively high sensitivity at high operating temperatures, which is of interest for high operating temperature sensor applications. The high responses for DCMS and RFMS deposited films are due to the formation of Pt, Pd and PdO catalytic layers, as well as a fine-grained structure with $D_g$ = 15 nm [60,61,64]. In most papers, $t_{res}$ and $t_{rec}$ reached several *min* and even tens of min [47,62,64]. The repeatability of the TiO$_2$ thin film characteristics when exposed to H$_2$ was not practically considered. For IBSD deposited TiO$_2$ thin films annealed at $T_{ann}$ = 1000 °C with $d$ = 130 nm the lowest $t_{res}$ and $t_{rec}$ were 23.4 s and 108.6 s, respectively.

IBSD deposited TiO$_2$ thin films demonstrated the highest response to NO$_2$. In references [65,66], TiO$_2$ thin films showed sensitivity to gas at RT. However, the $S$ was low, $t_{res}$ and $t_{rec}$ were hundreds of s. In ref [66] the authors analyzed the effect of NO$_2$ on frequency properties of TiO$_2$ thin films. The frequency response was low.

It can be concluded that IBSD deposited TiO$_2$ thin films are of interest for high operating temperature sensor applications. At the same time, in most of the research TiO$_2$ thin films were modified with additives or irradiated with UV. The characteristics of such films and IBSD deposited TiO$_2$ thin films are comparable in most cases. The capabilities of IBSD for the modification of TiO$_2$ thin films [33,37] may allow the achieving of superior performance for sensors in the future.

### 3.5. Sensory Effect

Estimates for TiO$_2$ thin films with a rutile structure showed that in the range of $T_{oper}$ = 300–750 °C, the ratio $L_D > D_g/2$ took place, where $L_D$ is the Debye length. The possible inclusion of the anatase phase in TiO$_2$ films annealed at $T_{ann}$ = 800 °C did not significantly affect the ratio between $L_D$ and $D_g/2$. Thus, the transport of charge carriers in TiO$_2$ films was not affected by the presence of a potential barrier at the boundaries of small grains and large agglomerates that formed after annealing of the films. This is typical for IBSD deposited MOS films, as seen in Ref. [39].

The power index for films at $T_{oper}$ = 750 °C $l$ = 0.20–0.23 (Table 4). The values of $l$ at other identical conditions were determined by the film surface microrelief [67,68], which changed with varying annealing conditions. It was shown in ref [53] that $l = \frac{1}{4}$, 1/5 and 1/6 occurred at $T_{oper}$ > 800 °C and were characteristic for the interaction of oxygen molecules with TiO$_2$ bulk defects, namely, oxygen vacancies and interstitial Ti atoms. A significant contribution to the gas sensitivity of TiO$_2$ films, concerning the interaction of gas molecules, oxygen vacancies and other bulk defects at $T_{oper} \leq 750$ °C, could be neglected due to diffusion limitations [53–55]. The conductivity of the film changes as a result of chemisorption of gas molecules on the semiconductor surface. Oxygen molecules are chemisorbed on the film surface in an air atmosphere. Oxygen captures electrons from the conduction band of the semiconductor and forms a layer depleted in charge carriers on the film pre-surface region. Oxygen is chemisorbed on the semiconductor surface in the molecular O$_2{}^-$(c) and atomic O$^-$(c), O$_2{}^-$(c) forms [69]. The molecular form of chemisorbed oxygen dominated at $T_{oper}$ < 150 °C. With a further increase in $T$, dissociative adsorption of oxygen molecules took place and the predominant forms of chemisorbed oxygen were O$^-$(c) and O$_2{}^-$(c). The obtained values of $l$ ~0.25 indicated the predominance of O$_2{}^-$(c) on the surface of TiO$_2$ thin films. It is worth noting that the O$^-$(c) form was the most reactive. A negative charge on the surface of the $n$-type film led to the upward bending of energy bands $eV_s$, where vs. is the surface potential, and $e$ is the electron charge. In the value $eV_s \sim N_i{}^2$, $N_i$ is the surface density of chemisorbed oxygen ions. The mechanism of the sensory effect described in ref [39] could be used for the IBSD TiO$_2$ thin films. The total

conductance of the $TiO_2$ film was $G_t = G_b + G_s$, where $G_b$ is the bulk conductance; $G_s$ is the surface conductance. The value of $G_s$ depends on the chemisorption of gas molecules on the semiconductor surface.

The expression describing the relationship between $G_t$ and vs. for an *n*-type semiconductor at vs. > 0 has the following form ref [39]:

$$G_t = G_b \times [1 - (L_D/d) \times [eV_s/(kT)]] \tag{6}$$

It can be noted from expression (6) that an increase in the oxygen concentration in the chamber with $TiO_2$ films led to a decrease in total conductance. Expression (6) took place at $L_D/d \ll 1$ and a small band bending $eV_s/(kT) \ll 1$. According to our estimates these inequalities were valid for our experimental conditions.

Interactions of previously chemisorbed $O^-_{(c)}$ and molecules of reducing gases can be represented in the following forms:

$$\begin{aligned} H_2 + O^-_{(c)} &\to H_2O + e; \\ CO + O^-_{(c)} &\to CO_2 + e, \\ CH_4 + 4O^-_{(c)} &\to CO_2 + 2H_2O + 4e. \end{aligned} \tag{7}$$

As a result of these reactions $N_i$, and $eV_s$ decrease, and electrons return to the conduction band of semiconductors. The reaction products between reducing gases and $O^-_{(c)}$ are desorbed as neutral $CO_2$ and $H_2O$ molecules. The following reactions can occur on the semiconductor surface when exposed to oxidizing gases $CO_2$, $NO_2$, and $NO$, [70,71]:

$$\begin{aligned} NO_2 + S_a + e^- &\to NO_2^-, \\ NO_2 + O^-_{(c)} &\to NO_3^-, \\ NO + S_a + e^- &\to NO^-, \\ NO + O^-_{(c)} &\to NO_2^-, \\ CO_2 + S_a + e^- &\to CO_2^-. \end{aligned} \tag{8}$$

Molecules of oxidizing gases can be chemisorbed onto the free adsorption center $S_a$ without the interaction with $O^-_c$ ions capturing electrons from the conduction band of the semiconductor. In mixtures of air + $NO_2$ $eV_s \sim (N_{iA} + N_{NO2})^2$, air + $NO$ $eV_s \sim (N_{iA} + N_{NO})^2$ and air + $CO_2$ $eV_s \sim (N_{iA} + N_{CO2})^2$, where $N_{iA}$ is the surface density of chemisorbed oxygen ions in the air atmosphere; $N_{NO2}$, $N_{NO}$ and $N_{CO2}$ are the surface densities of chemisorbed $NO_2^-$, $NO^-$ & $CO_2^-$-ions, respectively. An additional negative charge on the surface of $TiO_2$ films leads to a greater increase in $eV_s$ and, consequently, to a decrease in their $G_t$. Equations (7) and (8) are the simplest possible, and fundamentally explain the observed sensory effect. Many reasonable variants of other reactions between gas molecules and previously chemisorbed $O^-$ ions on the surface of metal oxide semiconductors have been proposed.

The interaction of gas molecules with $O_2^-_{(c)}$ is not considered in detail in the literature. The authors in [72–74] consider that the processes of interactions of gas molecules with $O^-_{(c)}$ and $O_2^-_{(c)}$ are similar, but the corresponding reactions are not given. It can be assumed that there is a step-by-step interaction of gas molecules with $O_2^-_{(c)}$. At high operating temperatures $O_2^-_{(c)}$ ions are predominant and a dissociative adsorption of gas molecules occurs on the semiconductor surface. In the example of $H_2$, a similar process can be represented as follows:

$$\begin{aligned} H_2 &\to 2H, \\ H + O_2^-_{(c)} &\to OH^-_{(c)} + e; \\ OH^-_{(c)} &\to OH + e. \end{aligned} \tag{9}$$

After the dissociation of the molecule, the atomic hydrogen H interacts with the $O_2^-_{(c)}$ ion, resulting in formation of a hydroxyl group $OH^-_{(c)}$ on the semiconductor surface with a localized electron, and an electron enters the $TiO_2$ conduction band. The $OH^-_{(c)}$ groups neutralize and desorb on the semiconductor surface [75].

## 4. Conclusions

The structural, electrically conductive, and gas-sensitive properties of $TiO_2$ films, with thicknesses of 130 nm and 463 nm, synthesized by the IBSD method, and subjected to high-temperature annealing in air were investigated. The IBSD $TiO_2$ films annealed at 800 °C and 1000 °C belong to the rutile phase of $TiO_2$. The electrically conductive properties of $TiO_2$ thin films in dry pure air are similar to those of bulk samples obtained by melt and high-temperature methods. The effect of $H_2$, CO, $CO_2$, $NO_2$, NO, $CH_4$ and $O_2$ on the electrically conductive properties of $TiO_2$ thin films in the operating temperature range of 200–750 °C was studied. The prospects of $TiO_2$ films deposited by IBSD for the development of high temperature operating $O_2$ sensors were demonstrated. The operating temperature of the maximum response to $O_2$ corresponded to 750 °C. $TiO_2$ films with a thickness of 130 nm and annealed at 800 °C manifested the highest response to $O_2$, which was 7.5 arb. un. when exposed to 40 vol. %. An increase in the annealing temperature up to 1000 °C at the same film thickness made it possible to reduce the response and recovery times by 2 times, due to changes appearing in the microstructure of the film surface. The films exhibited high responses to $H_2$ and $NO_2$ at an operating temperature of 600 °C and the largest sensitivity to nitrogen-containing oxides. An appropriate mechanism of the sensory effect in IBSD $TiO_2$ thin films was proposed. The great significance of various atomic forms of chemisorbed oxygen on the semiconductor surface was revealed.

**Author Contributions:** Conceptualization, A.V.A.; methodology, B.O.K., V.V.K., V.A.N., M.M.Z., N.N.Y. (Nikolay N. Yudin) and A.V.C.; software, V.V.K.; validation, A.V.A. and N.N.Y. (Nikita N. Yakovlev); formal analysis, A.V.A., N.N.Y. (Nikita N. Yakovlev), V.V.K. and V.A.N.; investigation, N.N.Y. (Nikita N. Yakovlev), B.O.K., V.V.K., V.A.N., M.M.Z., N.N.Y. (Nikolay N. Yudin), S.N.P., N.N.E., A.V.C. and M.P.S.; resources, M.M.Z., N.N.E. and A.S.O.; data curation, A.V.A. and N.N.Y. (Nikita N. Yakovlev); writing—original draft preparation, H.B.; writing—review and editing, A.V.A.; visualization, N.N.Y. (Nikita N. Yakovlev); supervision, A.V.A.; project administration, A.V.A.; funding acquisition, A.V.A. All authors have read and agreed to the published version of the manuscript.

**Funding:** The research was carried out with the support of a grant under the Decree of the Government of the Russian Federation No. 220 of 9 April 2010 (Agreement No. 075-15-2022-1132 of 1 July 2022) and Ministry of Science and Higher Education of the Russian Federation, project No. FSWM-2020-0038. Studies of the deposition of $TiO_2$ films by the IBS were supported by a grant from the Russian Science Foundation No. 22-22-20103 (https://rscf.ru/project/22-22-20103/ (accessed on 1 September 2022)) and funds of the Tomsk Region Administration.

**Institutional Review Board Statement:** Not applicable.

**Informed Consent Statement:** Not applicable.

**Data Availability Statement:** All data that support the findings of this study are included within the article.

**Conflicts of Interest:** The authors declare no conflict of interest.

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
