# Peer review of "Gas Sensitivity of IBSD Deposited TiO2 Thin Films"

_coatings, doi:10.3390/coatings12101565_

Round 1

Reviewer 1 Report

The authors prepared thin films on a substrate using the ion beam sputter deposition method, evaluated their sensing performances against various gaseous, and discussed their sensing mechanisms. However, before reviewing the article, I would like to confirm some points relating to the characterization of the prepared films. The authors characterized the prepared films by the X-ray diffraction (XRD) method and concluded the film is TiO2 in the rutile phase. Please add the information on the X-ray source for the XRD measurements. I think the Cu Ka is often used as an X-ray source of a laboratory-scale X-ray diffractometer. I checked several papers reporting XRD patterns of rutile TiO2 powder, nanoparticles, and nanorods collected using Cu Ka X-ray, and found that they showed strong diffraction peaks at around 27, 36, 41, and 54 degrees corresponding (110), (101), (111), and (211), respectively (See the references, Chem. Lett., 2003, 32, 638-639; Chem. Mater., 2008, 20, 3435-3442; J. Phys. Chem. B, 2006, 110, 25366-25370; Appl. Surf. Sci., 2003, 212-213, 255-263; Sci. Rep., 2014, 4, 5769 (Supplementary Information)). However, these peaks are not observed in Figure 2. If the XRD patterns in Figure 2 were recorded using a different X-ray source (wavelength), the observed peak positions should be different from those recorded using Cu Ka. In any case, please add the simulation XRD patterns of TiO2 in the rutile and anatase phases for comparison with the experimental results.

In lines 88 and 256, the film thickness values are written as 462 and 436 nm, respectively. I think the correct value is 463 nm according to the main text.

Please define what Och is. (Eq. 7 and main text)

How about the repeatability and durability of the sensors?

Reviewer 2 Report

The manuscript under review is a very nice and comprehensive study on the structural, electrical, and gas-sensing properties of TiO2 thin films. The method of deposition selected – IBSD has advantages over the other PVD methods in terms of synthesis of thin films of better quality. This method has not been applied yet for deposition of TiO2 films and therefore, the authors have employed a new approach to the TiO2 thin film synthesis and explored the thin film properties in terms of utilization for gas-sensors of a number of environment and technology-relevant gases, such as H2, CO, CO2, NO2, NO, CH4 and O2. Moreover, the authors have given a detailed description of the mechanism of the sensing effect in the above-described thin films. In this respect the manuscript is novel and of high interest to the audience of the journal. The paper should be accepted for publication after a minor revision following the comments below.

In the Abstract introduce the Abbreviation IBSD. You may change the term “grown” with “deposited” or “synthesized” because growth is used in epitaxy.

In the Introduction change “its” with “theirs”.

In Results and Discussion section

In Fig. 2 please assign all peaks observed.

Compare the gas-sensing characteristics of the IBSD TiO2 thin films with those of the films deposed by different methods (CVD or PVD).

In the manuscript there is no data on the stability of the gas-sensing characteristics of the IBSD TiO2 thin films developed. The paper will only gain if you add such data.

Reviewer 3 Report

The manuscript of Amlaev et al. evaluates the gas sensitivity of ion beam sputter deposited TiO2 thin films. The work is nicely presented and structured, and especially the introduction provides a strong overview of the gas sensitivity of TiO2 thin films from literature. Nevertheless, there are several aspects which need to be improved, as these include:

1. Abstract, experimental and results and discussion should clearly differentiate between the temperature used for annealing the layers (500, 800, 1000 deg) and the temperature variation during characterization measurements. Fpr example, the annealing temperature of the layers (critical for the properties), is not mentioned in the abstract.

2. Also, would there be a difference in the layers' properties if annealing would be done at 600deg?

3. XRD graphs and discussion - all peaks in the spectra should be assigned. Authors mention specifically only 2 rutile peaks, and one anatase peak for 800deg, stating that the rest of the peaks belong to rutile or the substrate. Please attribute each peak in the XRD plot. Also, if peaks from the substrate are visible (namely, from the polycrystalline sapphire plates), would these peaks not be visible in the as-formed TiO2 layer (not annealed)?

4. the anatase peak for the layer annealed at 800 deg - please provide a magnification of the peak in inset. It seems that there are 2 peaks between 23-25 2theta degrees. The XRD for the 500deg sample might be necessary to observe if there is more anatase present and the corresponding peak positions.

5. Fig 5 - for thin films annealed at 1000 ℃ the response to O2 was measured in the range of 500 – 750 ℃, while for 800deg it was in between 300 – 750 ℃. Why the difference? Moreover, if for the 1000℃ annealed layers the maximum response was at 750 ℃, what would happen at 800 ℃?

Round 2

Reviewer 1 Report

The manuscript has been brushed up according to the comments.

Fig 1 is slightly hard to see because a lot of lines for the comparison of experimental data and the XRD of corundum Al2O3 overlap on graphs. It is comfortable for readers if the authors rearrange the placements of the graphs in Fig. 1.

Reviewer 3 Report

The authors have addressed all the comments of the reviewer and made significant improvement in the manuscript, with respect to the mentioned comments. I appreciate the effort the authors' put into the revision. Also, and  Figure 2 (XRD data) is very thorough and nicely presented now.